# Depletion of key gut bacteria predicts disrupted bile acid metabolism in inflammatory bowel disease

Daniel Peterson,[1] Christopher Weidenmaier,[1] Sonia Timberlake,[1] Rotem Gura Sadovsky[1]

**ABSTRACT** The gut microbiome plays a key role in bile acid (BA) metabolism, where a diversity of metabolic products contribute to human health and disease. In particular, Inflammatory Bowel Disease (IBD) is characterized by a low concentration of secondary bile acids (SBAs), whose transformation from primary bile acids (PBAs) is an essential function performed solely by gut bacteria. BA-transformation activity mediated by the bile acid inducible (bai) operon has been functionally characterized in the genus *Clostridium*, and homologous bai gene sequences have been found in metagenome-assembled genomes (MAGs) belonging to other taxa in the human gut, but it is unclear which species of bai-carrying bacteria perform physiologically significant amounts of bile acid transformation in healthy and sick individuals. Here, we analyzed hundreds of stool samples with paired metagenomic and metabolomic data from IBD patients and controls and found that the abundance of the bai operon in metagenomic samples was highly predictive of that sample's high- or low-SBA metabolic state. We further found that bai genes from the *Clostridium* species best characterized as BA transformers were more prevalent in IBD patients than in non-IBD controls, while bai genes from uncharacterized taxa known only from MAGs were much more physiologically relevant in non-IBD samples. These un-isolated clades of BA-transforming bacteria merit further research; as beyond their prevalence in the human population, we found some cases in which they engrafted in IBD patients who had undergone fecal microbiota transplantation and experienced a clinical response.

**IMPORTANCE** In this paper, we identify specific bacteria that perform an important metabolic function in the human gut and demonstrate that in the guts of a large subset of patients with IBD, these bacteria are missing and the function is defective. This is a rare example where the correlation between the absence of specific bacteria and the dysfunction of metabolism is directly observed, not in mice nor in the lab, but in physiologic microbial communities in the human gut. Our results point to a path for studying how a small but important set of bacteria is affected by conditions in the IBD gut and perhaps to the development of interventions to mitigate the loss of these bacteria in IBD.

**KEYWORDS** bile acids, bai, FMT, IBD, dysbiosis, gut microbiome

The gut microbiome plays a key role in bile acid (BA) metabolism, where a diversity of metabolic products contributes to human health and disease (1). Among the metabolic signatures that characterize Inflammatory Bowel Disease (IBD) is a disrupted BA pool composition (2). BA metabolism pathways modify BAs into dozens (and recent research suggests thousands) (3) of different molecules. The relative abundances of these molecules differ between IBD patients and healthy individuals. Specifically, in the two main subtypes of IBD, Ulcerative Colitis (UC) and Crohn's Disease (CD), the host-generated primary bile acids (PBAs) are elevated relative to the microbiome-modified

Address correspondence to Rotem Gura Sadovsky, rgurasadovsky@gmail.com.

The authors declare no conflict of interest.

secondary bile acids (SBAs) and their downstream metabolism products. The fact that this metabolic signature is measured both in stool (2, 4, 5) and in peripheral blood (6) suggests that the signature reflects real differences in systemic metabolite ratios, rather than artifacts such as malabsorption or transport of BAs, which are known perturbed processes in IBD (7). In fact, relative abundances of BA species have been suggested as candidates for diagnostic biomarkers, as these abundances are correlated not only with IBD diagnosis but also with disease activity and symptoms, both in adults (6) and in children (8).

SBAs and downstream SBA metabolites exert anti-inflammatory activities relevant to IBD, as indicated by *in vitro* and *in vivo* studies. For example, the SBA lithocholate (LCA) and its derivatives can suppress inflammatory cytokine release by gastrointestinal epithelial cells (9) and directly inhibit IBD-relevant T cell populations in pro-inflammatory states (5, 10, 11). In addition, LCA and deoxycholate (DCA) metabolites can expand gastrointestinal mucosal and peripheral regulatory T cells (11–13). These activities are likely mediated by activation of BA receptors like TGR5 and modulation of a key transcription factor involved in T cell development (RORγt). *In vivo* studies demonstrate the therapeutic potential of microbiome therapeutics with SBA-increasing activity or direct SBA supplementation in IBD models (14, 15).

Transformation of PBAs into SBAs has been studied biochemically and *in vivo* using model organisms from a single bacterial genus (*Clostridium*) (16), but it is not clear which bacteria actually perform this function in human populations. Known BA-transforming bacteria harbor the bile acid-inducible (bai) operon, which includes eight Bai enzymes (17). These enzymes catalyze the key transformation reaction, dehydroxylation at C7 of PBAs, which leads to the production of the main SBAs, LCA and DCA, from the substrates CDCA and CA, respectively (18). This process has mostly been studied in a single species, *Clostridium scindens* (19), but it has not been shown that this organism or related species are prevalent or abundant in human guts. In fact, bai operons have been found in many bacterial genomes and metagenome-assembled genomes (MAGs), and non-*Clostridium* MAGs with bai genes appear much more abundant than the canonical *Clostridium* BA transformers (20, 21). However, the actual contribution of these taxa (and the bai operon as a whole) to physiologically relevant levels of BA transformation in the human gut remains unclear though one recent study has shown a correlation between the metagenomic abundance of the baiE gene and the SBA:PBA ratio, as well as differences in the abundances of several baiE sequence clusters between IBD patients and control samples (22). It is still unknown whether variability in bai-carrying gut bacteria between healthy individuals and IBD patients explains the disrupted SBA:PBA ratio associated with IBD. Generally, the altered BA pool in IBD patients likely results from many factors, including disrupted host metabolism, uptake and regulation, and changes to microbiome metabolism. Understanding to what extent the SBA:PBA ratio is controlled by the gut microbiome is important when considering targeted microbiome therapies for IBD, where individual strains can be given to patients to augment the SBA deficiency.

Here, we analyzed hundreds of stool samples with paired metagenomic and metabolomic data and found that the host's SBA:PBA ratio was, indeed, explained, to a large extent, by the abundance of bai genes in the gut community. Investigating the sequence diversity of bai genes in these metagenomes revealed that three bai-carrying taxa are much more prevalent and abundant than the rest, which suggests low redundancy of this biological function in the human gut, a finding consistent with the depletion of SBAs in dysbiotic IBD patients. We also found that the BA-transforming bacteria most abundant in IBD patients differed from those most abundant in non-IBD controls, with the single most abundant cluster of control-associated bacteria unrepresented in isolate collections. These unisolated BA-transforming bacteria hold potential for drug development, as we observed them engrafting in IBD patients who had undergone fecal microbiota transplantation and experienced a clinical response.

## MATERIALS AND METHODS

### Bile acid metabolomic data acquisition

We performed an exhaustive literature search for data sets with high-quality stool metagenomic sequencing and metabolomic measurements of IBD patients and controls. Two studies met our criteria for sequencing depth (more than 10 million reads per sample) and bile acid measurements (at least one estimate per pair: CA/CDCA and DCA/LCA). We obtained metabolomic measurements of fecal samples from the supplementary materials of Franzosa et al. (23) and Lloyd-Price et al. (2). These studies performed untargeted metabolomics, which provides ion abundance values that cannot be converted to absolute concentrations. We, therefore, defined a key bile acid transformation phenotype by calculating the ratio of measured deoxycholate (DCA) to cholate (CA) abundance. Lithocholate (LCA) measurements were not available in the Franzosa et al. data set, but DCA:CA ratio was highly correlated with LCA:CDCA ratio in the Lloyd-Price et al. samples, suggesting either metric could serve as an effective proxy for the overall ratio of SBAs-to-PBAs in the gut. Assuming similar ionization frequencies of DCA and CA, samples could then be characterized as high-secondary-bile-acid phenotype if they had more DCA than CA, or low-secondary-bile-acid phenotype if they had more CA than DCA. We then utilized a chi-square test to assess the association between bile acid phenotype and disease diagnosis with the python package scipy.stats.chi2_contingency (v1.10.0).

### Bai gene reference sequence collection

We based our reference set of bai gene amino acid sequences on the MAG search performed by Vital et al. (21). We supplemented this set of MAG-derived sequences with isolate-genome-derived bai sequences (in order to provide taxonomic waypoints and to identify clades that include isolated representatives) by downloading the MIDAS database of dereplicated isolate genomes (v1.2) (24) and mapping the Vital reference bai sequences against all the open reading frame (ORF) sequences in each MIDAS cluster's representative genome with diamond (v0.9.14). Any MIDAS ORF with at least 70% amino acid identity to a Vital reference sequence over 70% of the 90th quantile of alignment lengths for that gene was considered to be a reference bai sequence match. The MIDAS genomes with ORF matches to at least four of the eight bai gene families were added to our reference bai sequence set.

### Metagenomic estimates of bai operon abundance

Raw metagenomic sequencing reads from the Franzosa et al. and Lloyd-Price et al. human gut samples were downloaded from NCBI's sequencing read archive. We quality-filtered and trimmed the reads with trimmomatic (v0.38, parameters MINLEN:75 TRAILING:20 SLIDINGWINDOW:4:20) and removed any reads that mapped to the hg19 human genome reference with bowtie2 (v2.3.5, default parameters). To estimate the relative abundance of bai genes in each sample, we mapped the trimmed and filtered reads to our reference set of bai gene amino acid sequences. Reads were mapped with diamond (v0.9.14) and using a threshold of a minimum 90% identity over at least 25 amino acids, with the best single match assigned.

In each gut metagenome sample, we counted the number of reads mapping to each of the eight bai operon genes, ignoring which particular genome provided the reference sequence. Dividing these counts by the number of quality-filtered reads in the sample's data set provided a depth-normalized estimate of the relative abundance of each bai gene in each sample. To estimate the relative abundance of the bai operon overall, we took the geometric mean of the abundance across all eight genes. Genes with no reads mapped in a given sample were assigned a pseudocount of $10^{-9}$ before taking the geometric mean across the bai genes. Our reference set of bai gene sequences and code to calculate the above metrics for any input shotgun metagenomic sequencing samples are available on github (https://github.com/daniel-a-peterson/bai_operon_MG_abund).

We then assessed whether this metagenomic estimate of the abundance of the bai operon was predictive of the binary high-vs-low SBA phenotype in each data set using Mann-Whitney $U$ tests (scipy.stats.mannwhitneyu, v1.10.0). To summarize the relationship, we calculated the area under the curve (AUC) of the receiver operating characteristic curve by dividing the calculated U statistic by the product of the sample sizes in each BA phenotype class.

## Bai gene clustering and phylogenetic analysis

Next, we investigated the diversity, abundance, and prevalence of the bacterial taxa that contained bai genes in the Franzosa et al. samples. We first calculated the amino acid sequence identities between all pairs of genes within each bai family in our bai gene reference database with blastp (v2.7.1). We then averaged the percentage identity across the eight bai gene families (sum of matching amino acids divided by sum of alignment lengths) for all genome pairs, excluding those for which less than 80% of the total bai gene sequence length aligned. The reference genomes were subsequently clustered at a threshold of 90% amino acid identity across the bai operon. Using these cluster assignments for each bai reference genome, we determined the best-hit bai cluster for each mapped read from the metagenomes above and assessed the prevalence and abundance of each distinct cluster of bai-carrying bacteria. Prevalence was defined as the proportion of samples in which at least four bai genes from each cluster recruited at least one read. We also calculated correlations between the abundance of each pair of clusters with the Spearman rank-order correlation test (scipy.stats.spearmanr, v1.10.0).

To assess the phylogenetic relationships between the bacterial clusters, we identified one high-completeness (according to checkM) representative MAG from each species-level genome bin (95% whole-genome sequence identity cluster) identified in the Vital bai reference data set and the MIDAS bai-containing genomes. For each species-level representative, we identified 32 core single-copy genes from each genome using the MarkerScanner.pl software and hidden markov models provided by amphora2 (25). We then used mafft (v7.310) to align all sequences for each core gene individually, keeping only the longest sequence in the rare cases where one genome had multiple copies of that gene. Each alignment was trimmed with trimal (v1.4.1, -gappyout option), and then all alignments were concatenated. Finally, a maximum-likelihood phylogeny was inferred based on the concatenated core-gene alignment using RAxML (v8.2.10, GTRGAMMA model).

## Bai dynamics in an FMT data set

Finally, we assessed whether the abundance of bai genes can be influenced by fecal microbiota transplantation (FMT). We focused on a single FMT study (26) for which significant shotgun metagenomic sequencing reads from patient and donor fecal samples were available. These sequencing data were downloaded from NCBI's sequencing read archive, quality-filtered, and mapped against the bai gene reference database using the same parameters as above. We then assessed whether patients with clinical response to FMT (defined by Vaughn et al. as an increase in the patient's Harvey–Bradshaw Index of at least three without an increase in CD-related medications at 4 weeks post-FMT) were more likely to show increases in their abundance of bai operon genes using the Mann-Whitney $U$ test.

## RESULTS

### Low abundance of the bai operon predicts depletion of SBAs in stool

We discovered that in a subset of IBD patients, the fecal BA pool showed a deficiency in SBAs and that the gut microbiome's bai operon abundance largely explained that deficiency. Analyzing two independent multi-omics cohorts, we found that the abundance ratio of DCA to CA was bimodal, where in the majority of IBD samples that ratio was greater than 1, but in a large minority (30% of CD patients and 16% of UC

patients), it was smaller than 1, a state we termed SBA-deficient (Fig. 1). In contrast, only 4% of the non-IBD samples were SBA-deficient, suggesting this state is more common in IBD than in the individuals without IBD used as controls in these cohorts ($\chi^2$ = 20.73, $P$ = 3.15e−5, DOF = 2, $N$ = 283). Since certain gut bacteria are known to convert PBAs to SBAs using proteins encoded in the bai operon (17, 19), we checked whether SBA-deficiency correlates with the abundance of genes in that operon. To that end, we assembled a dereplicated reference set of 252 bai gene amino acid sequences (see Materials and Methods), mapped metagenomic reads to that reference set, and estimated the relative abundance of the bai operon overall by taking the geometric mean across all eight bai genes of the proportion of reads that map to each gene (see Materials and Methods for full description and code availability). Indeed, we found that the SBA-deficiency state, as assessed directly from metabolomics data, strongly associated with the abundance of the bai operon genes, as measured from metagenomics data generated from the same stool samples (Mann-Whitney test results for the two datasets were Franzosa 2018: $U$ = 5,253, $P$ = 3.38e−26, AUC = 0.966; Lloyd-Price: $U$ = 1,120, $P$ = 2.82e−7, AUC = 0.875). Strikingly, the bai gene abundance seemed to have a threshold effect on the DCA:CA ratio, where above a certain bai abundance, all samples contained a healthy-like BA balance (i.e., not SBA-deficient), while further increases in bai gene abundance had a negligible effect on the observed BA ratio. This observation suggested a minimum abundance of bai genes is required for the typical physiological conversion of bile acids in the gut, while an abundance higher than that minimum does not further explain the variance in the SBA to PBA ratio. Rather, the exact value of the ratio might depend on other factors such as the overall BA pool size and host regulation. This is consistent with a recent study in mice, where adding a low abundance of *Clostridium scindens* to a baseline community led to CA transformation into DCA (27).

Despite observing a similar positive relationship between the bai operon abundance and the BA balance in our two IBD case-control data sets, we also observed divergent patterns between these data sets in the samples with very low bai operon abundance (geometric mean relative abundance less than $10^{-8}$). In the Franzosa et al. data set, all samples with very low bai operon abundance also had low SBAs, but in the Lloyd-Price

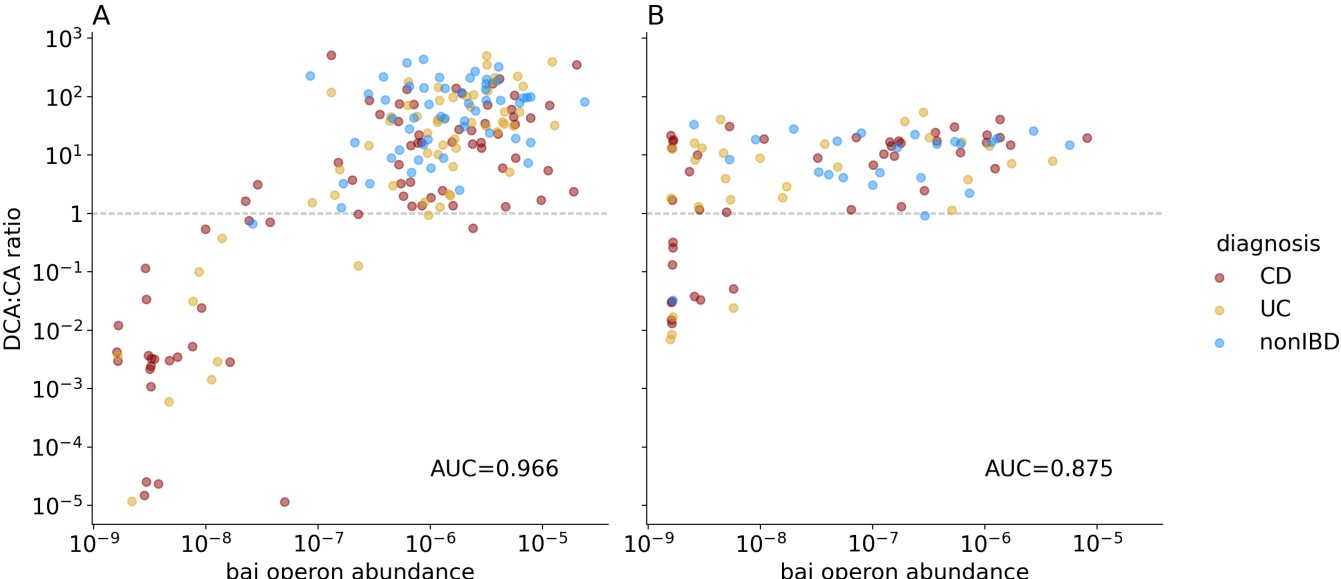

**FIG 1** The abundance of bai operon genes was strongly predictive of the abundance of SBAs in stool. Here, we show scatterplots of secondary:primary bile acid ratio against the relative abundance of the bai operon as estimated from metagenomic read mapping for samples from Franzosa et al. (23) (A) and Lloyd-Price et al. (2) (B). Bai operon abundance estimates were calculated as the geometric mean across all eight bai genes of the proportion of reads mapping to that gene family (see Materials and Methods). AUC values indicate the performance of bai operon abundance in predicting high vs low SBA concentration (DCA:CA ratio greater or less than 1; dashed line on plots) as calculated by Mann-Whitney $U$ test.

et al. data set, a substantial number of samples with very low bai operon abundance nevertheless displayed high levels of SBAs indistinguishable from the non-IBD group. It is possible that these high-SBA/low-bai-operon samples reflect the presence of BA-transforming genes that are sufficiently divergent from our reference bai genes to be undetected by our method. However, given the fact that both Lloyd-Price et al. and Franzosa et al. sampled the gut microbiomes of geographically similar populations, we believe it is more likely that methodological differences in DNA extraction and/or sequencing library preparation between the two studies led to different amplification and sequencing rate of the bai operon relative to the rest of the metagenome. In addition, the sequencing depth of the Franzosa et al. study appears sufficient to capture the functional cutoff for bai operon abundance to create high levels of SBAs, while the Lloyd-Price et al. study appears to lack sufficient sequencing sensitivity to detect bai genes in all samples with BA transformation occurring. The two studies also used different chromatography protocols for metabolomics data acquisition, which likely added to the divergence of our observations between the data sets. Given the strong predictive power we observed for bai genes in the Franzosa et al. samples, we focused our further analysis on that data set.

## A handful of bai-carrying taxa dominate the human gut community

Exploring the diversity of bai operon gene sequences, we found that three sequence clusters dominate our human gut samples, yet the most prevalent and abundant clusters differ between IBD patients and non-IBD controls. We clustered the reference bai operon sequences at 90% overall amino acid identity and obtained 17 clusters (see Materials and Methods). Of these clusters, only two were prevalent in non-IBD samples: cluster 2 (89% prevalence) and cluster 7 (72% prevalence; Fig. 2). No other cluster was detected in greater than 25% of non-IBD samples. In the CD patient samples, however, clusters 2 and 7 were much less prevalent, while cluster 10 was detected in more than half of

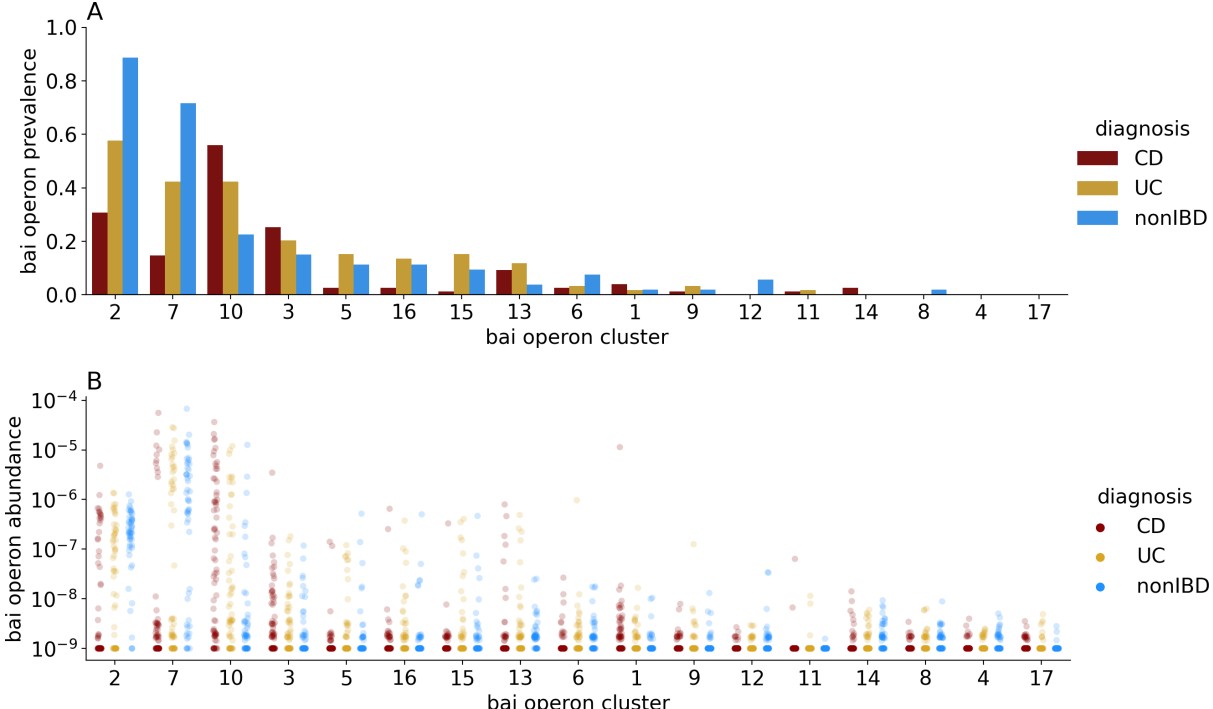

**FIG 2** Prevalence (A) and abundance (B) in Franzosa et al. samples of the 17 bai operon sequence clusters. Prevalence for each cluster was calculated as the proportion of samples in each diagnosis category in which at least four of the eight bai genes received at least one metagenomic read as a best-match hit. Bai operon abundance for each cluster was estimated by taking the geometric mean of the relative abundance across all eight bai genes, with zero-abundance genes given a $10^{-9}$ pseudo-count.

the samples (56% prevalence, Fig. 2). The UC patient samples showed an intermediate pattern, with clusters 2, 7, and 10 appearing in 40%–60% of samples. In addition to cluster prevalence, we analyzed the cluster abundance and found the three prevalent clusters above (2, 7, 10) were also the most abundant clusters. Cluster 7 had the highest mean abundance ($10^{-5.58}$) when it was detected in a sample, while cluster 2 was about an order of magnitude less abundant when detected (mean abundance $10^{-6.64}$) despite its higher prevalence. Cluster 10 was abundant in some IBD patients (and a few non-IBD samples when it was detected), approaching the abundance maximum of cluster 7, but its abundance distribution was much wider and its mean abundance when detected was $10^{-6.37}$. Lastly, correlation analysis showed clusters 2 and 7 were positively correlated and that cluster 10 was negatively correlated with both, suggesting two separate states of BA-transforming bacteria in the gut, which overlaps with the IBD status of the host (Fig. 3). In summary, we found that only a small minority of bai operon clusters were prevalent in our samples, with two clusters being prevalent and relatively abundant in healthy individuals and a single and different cluster being prevalent and relatively abundant in IBD patients.

Using a phylogenetic analysis, we found that bai operon clusters corresponded to mostly monophyletic clades of MAGs encompassing a small number of species-level genome bins (Fig. 4). Interestingly, the 17 bai clusters were divided into two major clades, one of which was composed of the large MAG-derived bai cluster 7 plus nine other clusters represented entirely by MAGs, without a single isolate genome. The other main clade was dominated by isolate genomes from the genus *Clostridium* (including cluster 10, which includes the canonical bai producer *C. scindens*), plus MAG clusters 2 and 13. Of the three aforementioned prevalent and abundant clusters 2, 7, and 10, only cluster 10 has any record of an isolated representative (using MIDAS, see Materials and Methods). The isolated strains whose bai genes match cluster 10 are all classified as *C. scindens*, including strains used in seminal work on BA transformation *in vitro* and *in vivo* (16). However, our analysis shows that *C. scindens* and the rest of cluster 10 are rarely found in healthy individuals, yet are prevalent in IBD patients. At the same time, the BA transforming bacteria most prevalent in healthy guts, i.e., clusters 2 and 7, have not been isolated, let alone characterized experimentally.

## One bai-carrying taxon engrafted in patients that responded to FMT treatment

Motivated by the opportunity to mitigate the depletion of SBAs and SBA-producing clades in IBD patients using therapeutic intervention, we sought evidence that supplementation of IBD patients with bai-carrying bacteria was possible by examining the abundance dynamics of these genes in a data set of microbiome samples from donors and patients undergoing FMT. In the Vaughn et al. (26) FMT data set, donors had

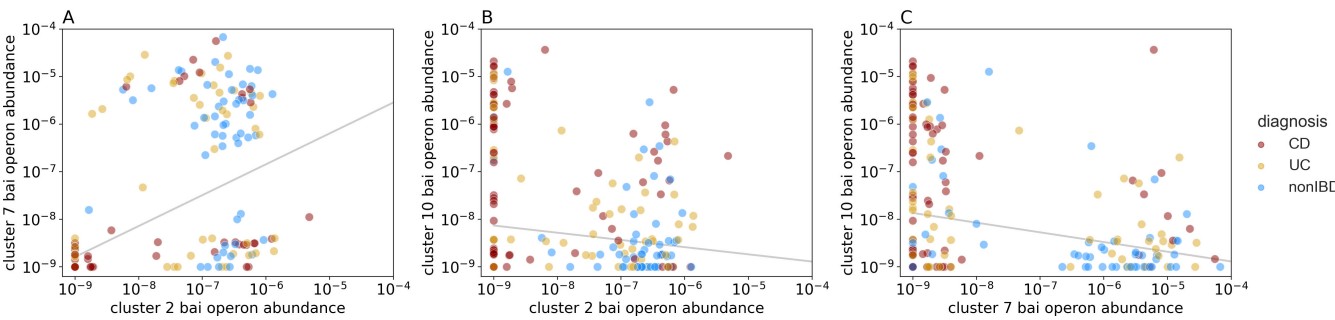

**FIG 3** Correlations between the three most prevalent bai operon clusters in the Franzosa et al. metagenomes, with lines representing the Theil-Sen median of all pairwise slopes and intercepts. (A) Despite the large phylogenetic distance between them, clusters 2 and 7 log10 abundances were positively correlated (Spearman's $\rho = 0.58$, $P = 1.49e{-}18$). (B) The abundances of clusters 10 and 2, on the other hand, were negatively correlated despite their phylogenetic proximity (Spearman's $\rho = -0.26$, $P = 2.88e{-}4$). (C) Clusters 10 and 7 were even more strongly negatively correlated (Spearman's $\rho = -0.35$, $P = 1.16e{-}6$), with very few samples showing high abundance of both clusters.

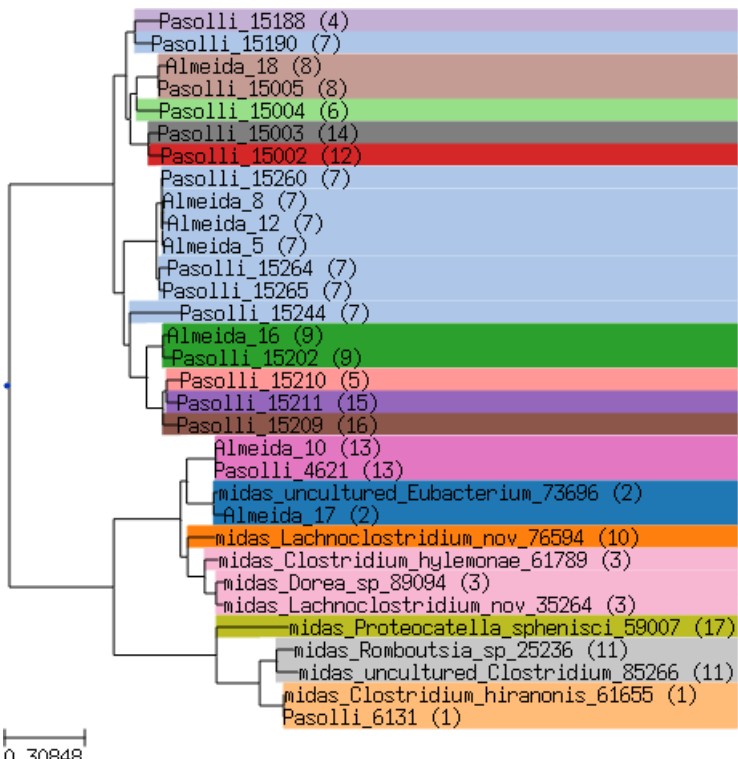

**FIG 4** Core-gene phylogeny of species-level genome bins from the Pasolli et al. (28) and Almeida et al. (29) MAG data sets found to carry the bai operon by Vital et al. (21), plus those representative isolate genomes from the MIDAS database with matches to the Vital et al. bai sequences for at least four of the eight bai genes. The numbers in parentheses and highlight colors indicate which bai operon cluster was found in each genome.

higher mean bai operon abundance than patients did before FMT. Moreover, patients that showed an increase in bai operon abundance after FMT were more likely to achieve clinical response (as defined by Vaughn et al.) than those with a decrease in bai operon abundance although this was not statistically significant in this small clinical study ($U$ = 42, $P$ = 0.12, AUC = 0.55, Fig. 5). Intriguingly, this association was driven by the low pre-FMT abundance and subsequent engraftment of bai cluster 7 taxa in a few patients (Fig. 6). Bai cluster 10, on the other hand, was relatively abundant in patients before FMT, tended not to change after FMT, and was not associated with clinical response. These FMT data show that bacteria from cluster 7 are the only BA-transforming bacteria that demonstrate the desirable properties of engraftment and potential association with clinical response to FMT in IBD and again highlight the opportunity in isolating and researching bacteria from that cluster. Bai gene sequences from this cluster and other bai clusters are included in our github repository (see Materials and Methods).

## DISCUSSION

Our results, building on previous work by Vital et al. (21) and Kim et al. (22), suggest that in most people, the essential function of transforming PBAs to SBAs is carried out by a handful of microbial taxa. This is surprising because important metabolic functions are generally highly redundant in the gut microbiome. For example, fermentation of plant fiber into butyrate, which nourishes colonocytes and contributes to immune signaling, is performed by well over a dozen bacterial species in a single individual's gut, across many families and even multiple phyla (30). Moreover, even BA metabolism pathways other than PBA transformation into SBA, i.e., deconjugation and isomerization, are prevalent across the gut microbiome (5, 31, 32). Yet, our results suggest that BA transformation

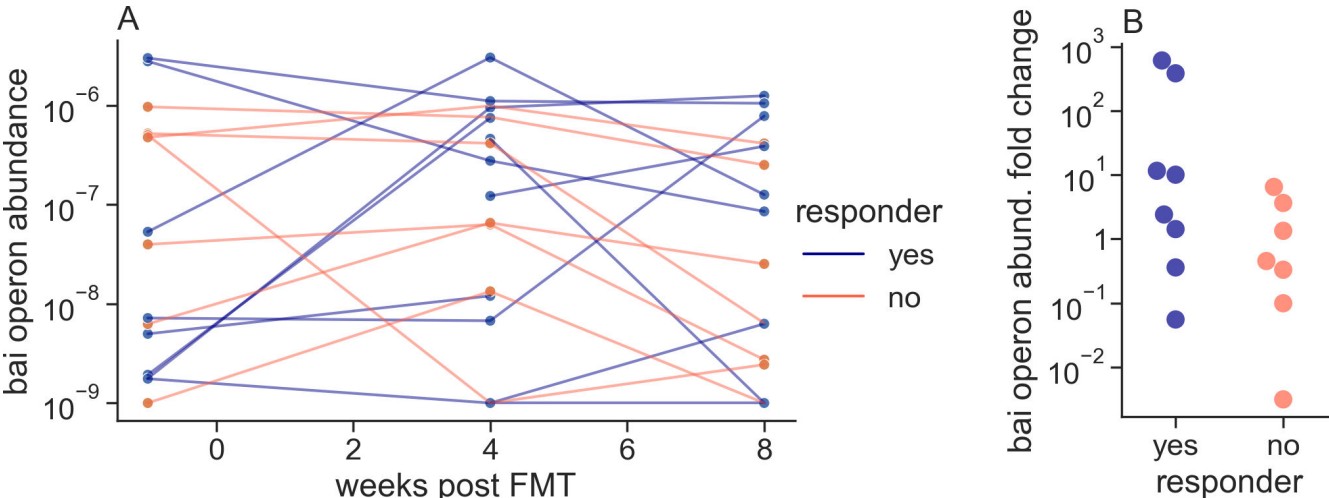

**FIG 5** (A) Abundance dynamics of the bai operon in Vaughn et al. (26) FMT recipients did not show a clear trend. (B) Nevertheless, patients who experienced an increase in bai operon abundance after FMT were more likely to achieve clinical response than those with a decrease in bai operon abundance although this association was not statistically significant ($U = 42$, $P = 0.12$, AUC = 0.55). Fold change of the bai operon abundance for each patient was calculated by dividing the geometric mean abundance of the bai operon in post-FMT samples by the geometric mean abundance of the bai operon in that patient's pre-FMT samples.

has minimal redundancy, carried out by three or fewer microbial clades in most gut microbiomes.

This low redundancy means that whenever the host gut environment does not support the growth of these specific microbes, their metabolic function is likely to be completely lost. It seems plausible that severe inflammation or other gut perturbations associated with IBD directly drive the few bai-carrying bacterial strains in many patients' guts to low abundance, thereby eliminating their metabolic function. However, it is difficult to establish causal direction in these situations, as the gut microbiome both affects and is affected by metabolite concentrations (33). A recent study made progress by showing that host T cell-driven inflammation in a graft-versus-host disease mouse model decreased the abundance of BA-metabolizing bacterial genes and altered the metabolomic BA pool composition although bai genes were not directly implicated (34). Predicting the bi-directional interactions of BA-transforming bacteria with human host physiology well enough to direct useful disease interventions will require understanding the ecology of the bai-carrying bacterial clades highlighted in our study on a species- and strain-specific level.

Our results highlight that BA transformation is predominantly performed by bacteria that have not been isolated. Of all bai-containing bacteria, the clade we annotated as cluster 7 is the most abundant and prevalent in healthy individuals and is rarely present in IBD patients. Moreover, in clinical FMT in IBD, cluster 7 is the only clade whose

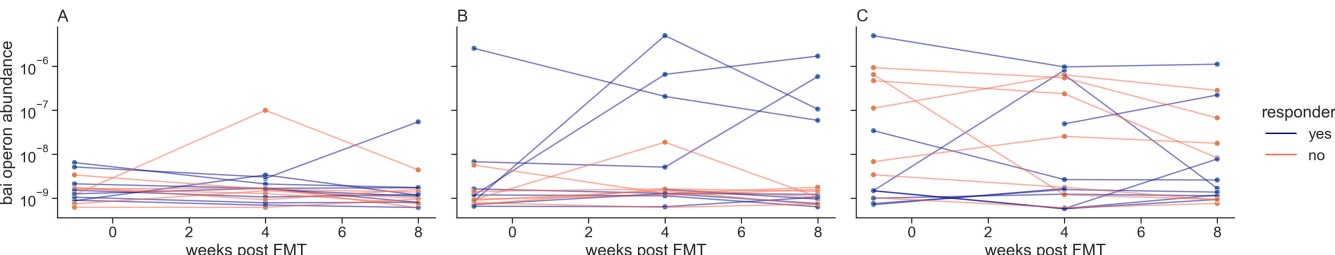

**FIG 6** Abundance dynamics in Vaughn et al. (26) FMT recipients of the three most prevalent bai operon clusters: cluster 2 (A), cluster 7 (B), and cluster 10 (C). Only cluster 7 shows a strong signature of engraftment in some patients (low abundance before FMT and high after). All four patients showing a high abundance of bai cluster 7 after FMT responded positively clinically to the FMT treatment, unlike most patients with a low abundance of cluster 7 after FMT. Some "jitter" was added to the y axis to separate lines clustered around $10^{-9}$ (pseudo-count for zero abundance).

engraftment correlates with clinical improvement. Yet, these bacteria have not been studied functionally and representative isolates from this clade are not available (21, 22). Instead, research on BA transformation usually focuses on *C. scindens*, which is rare in the healthy microbiome, prevalent in IBD patients, and tends not to engraft via FMT. Additional effort in isolating and studying bacteria from cluster 7 may shed light on the environmental conditions that support the growth of this key organism in our gut.

One of the observations in this work we find most intriguing is that cluster 10 (*Clostridium scindens*) is much more prevalent in IBD patients than in healthy individuals, corroborating previous findings (22). This distribution suggests that the IBD gut gives *C. scindens* a competitive advantage that it does not have in the healthy gut. Indeed, in the IBD gut, *C. scindens* is often the only detected BA-transforming bacterium. It remains unclear whether the presence of *C. scindens* is, therefore, beneficial to IBD patients by virtue of its BA metabolic activity, or if this trait is outweighed by other negative effects, for example, by competitively excluding the other BA-transforming bacterial taxa. Future research can shed light on this question by leveraging larger non-public data sets, such as the Lifelines initiative associated with the University of Groningen and the SPARC and RISK cohorts managed by the Crohn's and Colitis Foundation.

## AUTHOR AFFILIATION

[1]Finch Therapeutics, Somerville, Massachusetts, USA

## AUTHOR ORCIDs

Daniel Peterson ⓘ http://orcid.org/0000-0002-3024-3068
Rotem Gura Sadovsky ⓘ http://orcid.org/0009-0003-0663-5955

## AUTHOR CONTRIBUTIONS

Christopher Weidenmaier, Conceptualization, Investigation, Methodology, Writing – original draft, Writing – review and editing | Sonia Timberlake, Conceptualization, Investigation, Methodology, Validation, Writing – original draft, Writing – review and editing | Rotem Gura Sadovsky, Conceptualization, Investigation, Methodology, Project administration, Validation, Writing – original draft, Writing – review and editing.

## ADDITIONAL FILES

The following material is available online.

### Open Peer Review

**PEER REVIEW HISTORY (review-history.pdf).** An accounting of the reviewer comments and feedback.

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
