## [Reviewer comments · Microbiology Spectrum]

Microbiology Spectrum

Depletion of key gut bacteria predicts disrupted bile acid metabolism in inflammatory bowel disease

Daniel Peterson, Christopher Weidenmaier, Sonia Timberlake, and Rotem Gura Sadovsky

Corresponding Author(s): Rotem Gura Sadovsky, Finch Therapeutics

Review Timeline:

Submission Date:	August 10, 2024
Editorial Decision:	September 9, 2024
Revision Received:	November 13, 2024
Accepted:	November 15, 2024

Editor: Jan Claesen

Reviewer(s): The reviewers have opted to remain anonymous.

Transaction Report:

DOI: <https://doi.org/10.1128/spectrum.01999-24>

Re: Spectrum01999-24 (Depletion of key gut bacteria predicts disrupted bile acid metabolism in inflammatory bowel disease)

Dear Dr. Rotem Gura Sadovsky:

Thank you for the privilege of reviewing your work. Below you will find my comments, instructions from the Spectrum editorial office, and the reviewer comments.

Thank you for submitting your research to Spectrum. Your paper has now been evaluated by two independent Reviewers who are enthusiastic about your work (as am I)! The Reviewers have highlighted some comments and suggestions to help improve the manuscript (copied in this message below), and I would be happy to consider a revised version that addresses these in a point-by-point manner. Note that it is Spectrum policy to not use novelty as a decision criterion (and hence comments pertaining to this should not be addressed by additional experiments), and we do not necessarily request additional experiments/data is typically not required but can be included if available to address reviewer comments.

Revision Guidelines

Sincerely,
Jan Claesen
Editor
Microbiology Spectrum

Reviewer #1 (Comments for the Author):

The study by Peterson et al. identified the correlation between the presence of specific gut microbiota that metabolize primary bile acids into secondary bile acids and the ratio of secondary to primary bile acids in the gut microbiome of between healthy control and IBD patients. Through this, authors described uncultivated gut microbes that are thought to play a role in alleviation of IBD. There are some issues to be resolved for a publication. In addition, the manuscript should be rephrased more clearly and contain many grammatical errors.

1. The distribution and abundance patterns of both the well-known and newly highlighted clusters of the bai operon, as well as the metabolite levels and their prevalence in IBD, have already been addressed in previous studies (notably Kim et al., 2022). Therefore, this does not seem to present novel findings. The relationship between IBD and bile acids should be further elucidated in more depth in this paper (e.g., using transcriptome data).
2. The analysis related to FMT, particularly examining the significance between the abundance of secondary bile acid-producing gut microbes and recipient responders in IBD, has the potential to provide new insights. Expanding the metagenome data analysis following FMT in IBD patients could yield further clues. Additionally, it would be advisable to verify the statistical significance of responders for each bai operon cluster.
3. "The sequencing depth of the Franzosa study appears sufficient to capture the functional cutoff for bai operon abundance to create high levels of SBAs, while the Lloyd-Price study unfortunately appears to not have enough sequencing sensitivity to detect bai genes in all samples with bile acid transformation occurring. For this reason, we focused our further analysis on the Franzosa dataset."
 The differences in metagenome analysis methods, even between different studies, do not seem likely to cause significant discrepancies. However, the trend in the results may not be clearly evident due to differences in the methods used to measure metabolites. Standardizing the substance detection methods as much as possible between the two studies might lead to better results.
4. "Using a phylogenetic analysis, we found that bai operon clusters corresponded to mostly monophyletic clades of MAGs encompassing a small number of species-level genome bins (Fig 4)."
-> If the MAGs for each cluster containing the bai operon have been secured, analyzing and providing their taxonomic lineage would be more effective in offering insights into bile acid-metabolizing microbes in the human gut. This could be achieved using genome-based identification tools such as EZBioCloud and GTDB-tk.
5. This paper does not include subtitles for each section. Adding subtitles could help organize the content more systematically.
6. It appears that only a log scale was applied to the mapping values when measuring abundance in the methods section. Normalization of these values against the total read count is necessary. Additionally, since the bai operon consists of 6-8 genes, a detailed explanation of the method used to assess abundance is required.

Reviewer #2 (Comments for the Author):

In the present manuscript titled „ Depletion of key gut bacteria predicts disrupted bile acid metabolism in inflammatory bowel disease" Peterson and colleagues examined bacteria converting bile acids in IBD patients and healthy controls based on in silico analyses of previously published data.

Overall, the study addresses an important question and methods applied are generally sound. I have a few major suggestions that should be addressed.

>The values of bai relative abundances are difficult to interpret and seem to be very low - authors should represent results as % of communities (reads). At least give an example of calculations e.g. abundance of $10e-7$ refers to mean x average reads per bai gene for a sequencing depth of 2×10^7 . This will allow comparing results to existing literature. The same applies for DCA:CA ratio.

>Results are largely based on a single dataset - is this the only one available? If true, authors should state that in the manuscript.

>Results for the reference database should be given - what is the difference/benefit by including midas genomes compared with the Vital et al database?

>Figure 1

X Next to group-wise stratifications, I suggest to give also regression results along with lines in the plots.

X Separate analyses between patients and healthy might be good, as the signal is largely dominated by patient samples with low in bai abundance. In healthy individuals I feel no correlation between bai abundance and DCA:CA ratio was detected?

>Figure 2

X I suggest to deduce differences between CD and UC as they seem to harbor different communities and the manuscript would benefit from such insights.

X Panel B should contain box-plots along with jitter plots in order to assess how many "0s" are present for individual clusters.

>Figure 3

X I suggest to show regression lines on plots.

X In general, the information provided is of minor significance given that results are largely dominated by host status (patient vs healthy), which is already shown in Fig 2.

>Figure 5 and 6

X how is "response" defined?

X next to lines, actual data points should be given.

Reviewer #1 (Comments for the Author):

The study by Peterson et al. identified the correlation between the presence of specific gut microbiota that metabolize primary bile acids into secondary bile acids and the ratio of secondary to primary bile acids in the gut microbiome of between healthy control and IBD patients. Through this, authors described uncultivated gut microbes that are thought to play a role in alleviation of IBD. There are some issues to be resolved for a publication. In addition, the manuscript should be rephrased more clearly and contain many grammatical errors.

1. The distribution and abundance patterns of both the well-known and newly highlighted clusters of the bai operon, as well as the metabolite levels and their prevalence in IBD, have already been addressed in previous studies (notably Kim et al., 2022). Therefore, this does not seem to present novel findings. The relationship between IBD and bile acids should be further elucidated in more depth in this paper (e.g., using transcriptome data).

> We agree with the reviewer that a correlation between the abundance of bai genes and bile acid levels has already been discovered and published by Kim et al. 2022 and others we are citing. We reviewed those previous findings in the introduction, added references in the discussion, and highlighted our novel contributions:

- a) Secondary-to-primary bile acid ratios are distributed bi-modally, which divides human gut samples to an either SBA-normal or SBA-deficient phenotype.
- b) Bai gene abundance as measured across the whole operon predicts the binary phenotype described above with striking accuracy.
- c) Few bacterial taxa that carry the bai operon are prevalent in the human gut, which suggests the capacity of the microbiome to transform PBAs into SBAs relies on a handful of taxa.
- d) The most abundant of these taxa are undetected in IBD patients, which suggests that BA transformation in IBD relies more heavily on taxa that are less abundant and prevalent in individuals without IBD.

We further agree that additional data types, such as transcriptomics, can reveal more about the relationship between IBD and bile acids. In this work, we limited our scope to asking whether bile acid levels are explained by the presence of bai-carrying bacteria, whether those bacteria differ between healthy individuals and IBD patients, and whether engraftment of those bacteria correlate with clinical effects.

2. The analysis related to FMT, particularly examining the significance between the abundance of secondary bile acid-producing gut microbes and recipient responders in IBD, has the potential to provide new insights. Expanding the metagenome data analysis following FMT in IBD patients could yield further clues. Additionally, it would be advisable to verify the statistical significance of responders for each bai operon cluster.

> We agree with the reviewer that additional FMT datasets can deepen our understanding of bai-carrying bacteria in IBD patients and their effect on BA transformation. We hope that in the future, additional FMT datasets suitable for such analysis will be generated. In our experience, most FMT datasets aren't suitable for the analysis we developed, either because metagenomic data was not acquired or because the data was sampled at a depth that is insufficient for detection of low abundance genes like bai. For this reason, we analyzed a single high-quality FMT dataset, though the insight we derived from it has low statistical

power due to the small number of patients and donors. Nevertheless, the conclusions of our work rely most heavily on the cross-sectional datasets discussed in the first parts of the manuscript, while insight from the FMT dataset serves to add nuance to the importance of the different bai-carrying taxa in IBD.

3. *"The sequencing depth of the Franzosa study appears sufficient to capture the functional cutoff for bai operon abundance to create high levels of SBAs, while the Lloyd-Price study unfortunately appears to not have enough sequencing sensitivity to detect bai genes in all samples with bile acid transformation occurring. For this reason, we focused our further analysis on the Franzosa dataset."*

 *The differences in metagenome analysis methods, even between different studies, do not seem likely to cause significant discrepancies. However, the trend in the results may not be clearly evident due to differences in the methods used to measure metabolites. Standardizing the substance detection methods as much as possible between the two studies might lead to better results.*

> We agree that differences between the metabolomics methods used in these two studies may contribute to the differences we are seeing in the relationship between estimated bai operon abundance and secondary:primary bile acid ratio. The most prominent differences are in the chromatography protocols, which we cannot correct for once the data has been generated. We have added this point to the manuscript text. In addition to differences in metabolomics methods, the two datasets do differ in DNA extraction protocols, which is a known cause for discrepancies between datasets. We too would like to see increased standardization of metabolomic and metagenomic protocols (and more collection of those data in the first place!).

4. *"Using a phylogenetic analysis, we found that bai operon clusters corresponded to mostly monophyletic clades of MAGs encompassing a small number of species-level genome bins (Fig 4)."*

-> *If the MAGs for each cluster containing the bai operon have been secured, analyzing and providing their taxonomic lineage would be more effective in offering insights into bile acid-metabolizing microbes in the human gut. This could be achieved using genome-based identification tools such as EZBioCloud and GTDB-tk.*

> The taxonomic precision of the most abundant and important MAGs remains limited due to insufficient study and unavailable cultures, so we felt a basic phylogenetic approach best served the purposes of this study. Kim et al. (2022) do provide a thorough exploration of both phylogeny and taxonomy of bai-carrying taxa in the human gut (although they did not publish their sequences, so it was impossible for us to match our cluster IDs to their latin names). We would absolutely encourage further work to characterize the taxonomic, functional and ecological attributes of these bai-carrying MAGs, and we hope that by providing these gene sequences and MAG identifiers to publically available data this publication will make such research easier.

5. *This paper does not include subtitles for each section. Adding subtitles could help organize the content more systematically.*

> We have added subsection titles for the Methods and Results.

6. *It appears that only a log scale was applied to the mapping values when measuring abundance in the methods section. Normalization of these values against the total read count is necessary. Additionally, since the bai operon consists of 6-8 genes, a detailed explanation of the method used to assess abundance is required.*

> We thank the reviewer for the comment. The metric we used is depth-normalized, and takes the geometric mean of the relative abundance across all 8 bai genes. Paragraph 4 of the Methods section gives a step-by-step explanation of how our metric is calculated (and the code to implement it is also available in our github repo). Following this comment, we have summarized the metric in the results section and clarified the figure captions.

Reviewer #2 (Comments for the Author):

In the present manuscript titled „ Depletion of key gut bacteria predicts disrupted bile acid metabolism in inflammatory bowel disease" Peterson and colleagues examined bacteria converting bile acids in IBD patients and healthy controls based on in silico analyses of previously published data.

Overall, the study addresses an important question and methods applied are generally sound.

I have a few major suggestions that should be addressed.

>The values of bai relative abundances are difficult to interpret and seem to be very low - authors should represent results as % of communities (reads). At least give an example of calculations e.g. abundance of $10e-7$ refers to mean x average reads per bai gene for a sequencing depth of 2×10^7 . This will allow comparing results to existing literature.

The same applies for DCA:CA ratio.

> We thank the reviewer for the note. Our abundance metric does reflect the proportion of reads mapping to the bai genes out of the total metagenomic reads, but we present it on a log scale, because the relative abundance of these genes is indeed very low. We agree with the reviewer that it is preferred to use metrics common in the literature, and indeed the relative abundance metric we chose is quite common (e.g. Das et al. 2019, Song et al. 2019). While we had described our metric in the Methods section, we now added a short description of this metric in the Results section as well and improved the figures. We also included Python code to enable the reader to analyze data using this metric or modify it as they wish.

The DCA:CA ratio too is a standard metabolite ratio metric, but we present it on a log scale to show the wide range of this ratio across samples. We have added an explanation of this metric in the Results section and visually in the figures.

>Results are largely based on a single dataset - is this the only one available? If true, authors should state that in the manuscript.

> We added a description of our dataset selection process to the beginning of the Methods section. The datasets we focused on were the best two datasets we could find at the time of conducting this analysis. It was surprising how few datasets were available with a sufficient number of IBD patients, high quality metabolomic data, and deep metagenomic sequencing. Hopefully these types of datasets will become more common as costs come down on the technology side.

>Results for the reference database should be given - what is the difference/benefit by including midas genomes compared with the Vital et al database?

> The MIDAS genomes were included mainly to provide taxonomic waypoints for bai clusters and to reveal which clusters included known, isolated bacterial strains. Given that all clusters contained MAGs,

the mapping and predictive power of the reference dataset was likely unchanged by inclusion of the MIDAS genomes. We added a clarification regarding this topic in the methods section.

>Figure 1

X Next to group-wise stratifications, I suggest to give also regression results along with lines in the plots.

> We believe that showing regression results would be confusing for the reader because our statistical analysis does not include regression. We believe that a Mann-Whitney U test of the relationship between bai gene abundance (continuous variable) and high or low secondary bile acids (binary variable / classification) provides the most appropriate statistical test for this relationship. Rather than looking at the correlation across the whole y axis equally, we feel that researchers and clinicians would be particularly interested in the power of a metagenomic bai abundance metric to predict whether an IBD patient has high (healthy-like) secondary bile acids or is qualitatively depleted of this important metabolite class. Linear regression, while familiar to many readers, treats the whole y axis equally, ignoring the fact that the biological significance of the difference between a DCA:CA metabolite ratio of, for example, 1:10 and 10:1 is much greater than that between 10:1 and 1000:1.

X Separate analyses between patients and healthy might be good, as the signal is largely dominated by patient samples with low in bai abundance. In healthy individuals I feel no correlation between bai abundance and DCA:CA ratio was detected?

>We agree with the reviewer's observation that only IBD patients have low secondary bile acids (and low bai abundance) in these datasets. However we see more value in running a single statistical test in this case, as the simplest model is that the relationship between bai gene abundance and bile acid ratio is the same in both groups. Put another way, if we sampled thousands more individuals without IBD, we might expect to find a few people with low bai gene abundance, and we would predict that they would also have low secondary bile acids, even if in these datasets we did not observe non-IBD individuals that demonstrate this phenotype.

>Figure 2

X I suggest to deduce differences between CD and UC as they seem to harbor different communities and the manuscript would benefit from such insights.

> We did look at this, and there are some differences between the patient groups, but they are relatively subtle compared to the major findings we report. We therefore omitted them for clarity and to focus the paper on our main results. However, we agree this could be a fruitful area for further research.

X Panel B should contain box-plots along with jitter plots in order to assess how many "0s" are present for individual clusters.

> We agree that the proportion of zero-abundance samples for each cluster is an important metric, but we believe that Panel A of this plot, which explicitly focuses on the detection rate and prevalence of each cluster, conveys this information more effectively than a boxplot of abundances would. We also prefer the unobstructed presentation of the datapoint distribution without the boxplots. We have edited this figure caption for clarity.

>Figure 3

X I suggest to show regression lines on plots.

> We thank the reviewer for the suggestion, and have added regression lines to these plots.

X In general, the information provided is of minor significance given that results are largely dominated by host status (patient vs healthy), which is already shown in Fig 2.

> We respectfully disagree with the reviewer and feel that this figure does show interesting patterns of co-abundance, both across and within patient diagnosis groups. We do agree some of the effect is driven by differences in diagnosis groups, but trends within each group appear to match the overall pattern.

> *Figure 5 and 6*

X how is "response" defined?

> Clinical response is defined in the Vaughn et al. (2016) study as an increase in the patient's Harvey–Bradshaw Index of at least 3 without an increase in CD-related medications at 4 weeks post-FMT. We have clarified this point in the Methods and Results.

X next to lines, actual data points should be given.

> We have added dots at each data point.

Literature Cited

Das, Promi, Simonas Marcišauskas, Boyang Ji, and Jens Nielsen. 2019. "Metagenomic Analysis of Bile Salt Biotransformation in the Human Gut Microbiome." *BMC Genomics* 20 (1): 517.

<https://doi.org/10.1186/s12864-019-5899-3>.

Song, Ziwei, Yuanyuan Cai, Xingzhen Lao, Xue Wang, Xiaoxuan Lin, Yingyun Cui, Praveen Kumar Kalavagunta, et al. 2019. "Taxonomic Profiling and Populational Patterns of Bacterial Bile Salt Hydrolase (BSH) Genes Based on Worldwide Human Gut Microbiome." *Microbiome* 7 (1): 9.

<https://doi.org/10.1186/s40168-019-0628-3>.

Re: Spectrum01999-24R1 (Depletion of key gut bacteria predicts disrupted bile acid metabolism in inflammatory bowel disease)

Dear Dr. Rotem Gura Sadovsky:

Thanks for addressing the Reviewers' comments. I would like to congratulate you on the acceptance of your paper for publication in Spectrum!

Your manuscript has been accepted, and I am forwarding it to the ASM production staff for publication. Your paper will first be checked to make sure all elements meet the technical requirements. ASM staff will contact you if anything needs to be revised before copyediting and production can begin. Otherwise, you will be notified when your proofs are ready to be viewed.

Sincerely,
Jan Claesen
Editor
Microbiology Spectrum